# Uniform Asymptotic Approximation Method with Pöschl–Teller Potential

Rui Pan [1], John Joseph Marchetta [2], Jamal Saeed [1], Gerald Cleaver [2], Bao-Fei Li [3,4], Anzhong Wang [1,*] and Tao Zhu [3,4]

1 GCAP-CASPER, Physics Department, Baylor University, Waco, TX 76798-7316, USA; rui_pan1@baylor.edu (R.P.); jamal_saeed1@baylor.edu (J.S.)
2 EUCOS-CASPER, Physics Department, Baylor University, Waco, TX 76798-7316, USA; john_marchetta1@baylor.edu (J.J.M.); Gerald_Cleaver@baylor.edu (G.C.)
3 Institute for Advanced Physics & Mathematics, Zhejiang University of Technology, Hangzhou 310032, China; libaofei@zjut.edu.cn (B.-F.L.); zhut05@zjut.edu.cn (T.Z.)
4 United Center for Gravitational Wave Physics (UCGWP), Zhejiang University of Technology, Hangzhou 310032, China
* Correspondence: anzhong_wang@baylor.edu

**Abstract:** In this paper, we study analytical approximate solutions for second-order homogeneous differential equations with the existence of only two turning points (but without poles) by using the uniform asymptotic approximation (UAA) method. To be more concrete, we consider the Pöschl–Teller (PT) potential, for which analytical solutions are known. Depending on the values of the parameters involved in the PT potential, we find that the upper bounds of the errors of the approximate solutions in general are $\lesssim 0.15 \sim 10\%$ for the first-order approximation of the UAA method. The approximations can be easily extended to high orders, for which the errors are expected to be much smaller. Such obtained analytical solutions can be used to study cosmological perturbations in the framework of quantum cosmology as well as quasi-normal modes of black holes.

**Keywords:** loop quantum cosmology; cosmological perturbations; power spectrum; black holes; quasi-normal modes; gravitational waves





## 1. Introduction

A century after the first claim by Einstein that general relativity (GR) needs to be quantized, the unification of quantum mechanics and GR still remains an open question despite enormous efforts [1,2]. Such a theory is necessary not only for conceptual reasons but also for the understanding of fundamental issues, such as the big bang and black hole singularities. Various theories have been proposed, and among them, string/M-Theory and loop quantum gravity (LQG) have been extensively investigated [3–13]. Differences between the two approaches are described in [14,15].

LQG was initially based on a canonical approach to quantum gravity (QG) introduced earlier by Dirac, Bergmann, Wheeler, and DeWitt [16,17]. However, instead of using metrics as the quantized objects [16,17], LQG is formulated in terms of densitized triads and connections and is a non-perturbative and background-independent quantization of GR [18,19]. The gravitational sector is described by the SU(2)-valued Ashtekar connection and its associated conjugate momentum, the densitized triad, from which one defines the holonomy of Ashtekar's connection and the flux of the densitized triad. Then, one can construct the full kinematical Hilbert space in a rigorous and well-defined way [7–13]. An open question of LQG is its semiclassical limit: that is, are there solutions of LQG that closely approximate those of GR in the semiclassical limit?

Although the above question still remains open, concrete examples can be found in the context of loop quantum cosmology (LQC). (For recent reviews of LQC, see [20–30] and references therein.) Physical implications of LQC have also been studied using *the effective descriptions* of the quantum spacetimes derived from coherent states [31], whose validity has been verified numerically for various spacetimes [32–37], especially for states sharply peaked on classical trajectories at late times [38]. The effective dynamics provide a definitive answer to the resolution of the big bang singularity [39–48], which is replaced by a quantum bounce when the energy density of matter reaches a maximum value determined purely by the underlying quantum geometry.

To connect LQC with observations, cosmological perturbations in LQC have been also investigated intensively in the past decade, and a variety of different approaches to extend LQC to include cosmological perturbations have been developed. These include the dressed metric [49–51], hybrid [27,52–54], deformed algebra [55–58], and separate universe [59,60] approaches. For a brief review on each of these approaches, we refer readers to [29].

One of the major challenges in the study of cosmological perturbations in LQC is how to solve for the mode functions $\mu_k$ from the modified Mukhanov–Sasaki equation. So far, this has mainly been done numerically [20–30]. However, this is often required to be conducted with high-performance computational resources [61], which are not accessible to the general audience.

In the past decade, we have systematically developed the uniform asymptotic approximation (UAA) method initially proposed by Olver [62–64] and have applied it successfully to various circumstances [65–85] [1]. In this paper, we continue work on this by considering the case in which the effective potential has only zero points but without singularities. To be more concrete, we consider the Pöschl–Teller (PT) potential, for which analytical solutions are known [86]. The consideration of this potential is also motivated by the studies of cosmological perturbations in dressed metric and hybrid approaches [87,88], in which it was shown explicitly that the potentials for the mode functions can be well-approximated by the PT potential with different choices of the PT parameters. In particular, in the dressed metric approach, the mode function satisfies the following equation [87]

$$\mu_k''(\eta) + \left[k^2 - \mathscr{V}(\eta)\right]\mu_k(\eta) = 0, \tag{1}$$

in which $\mathscr{V}(\eta)$ serves as an effective potential. During the bouncing phase, it is given by

$$\mathscr{V}_{\text{dressed}}(\eta) \equiv \frac{\gamma_B m_{\text{Pl}}^2 (3 - \gamma_B t^2 / t_{\text{Pl}}^2)}{9(1 + \gamma_B t^2 / t_{\text{Pl}}^2)^{5/3}}, \tag{2}$$

where $\gamma_B$ is a constant introduced in [87], and $m_{\text{Pl}}$ and $t_{\text{Pl}}$ are, respectively, the Planck mass and time. This potential can be well-approximated by a PT potential

$$\mathscr{V}_{\text{PT}}(\eta) = \frac{\mathscr{V}_0}{\cosh^2 \alpha(\eta - \eta_B)}, \tag{3}$$

with

$$\mathscr{V}_0 = \frac{\gamma_B m_{\text{Pl}}^2}{3} = \frac{\alpha^2}{6}. \tag{4}$$

Here $\eta$ is the conformal time related to the cosmic time $t$ by $d\eta = dt/a(t)$. On the other hand, in the hybrid approach, the effective potential is given by

$$\mathscr{V}_{\text{Hybrid}}(\eta) = -\frac{\gamma_B m_{\text{Pl}}^2 (1 - \gamma_B t^2 / t_{\text{Pl}}^2)}{9(1 + \gamma_B t^2 / t_{\text{Pl}}^2)^{5/3}}, \tag{5}$$

which can be also modeled by the PT potential (3) but now with [88]

$$V_0 = \frac{m_{\text{Pl}}^2 \gamma_B}{9}, \quad \alpha^2 = \frac{2}{3} m_{\text{Pl}}^2 \gamma_B. \tag{6}$$

For more details, we refer readers to [87,88].

The rest of the paper is organized as follows: In Section 2, we provide a brief review of the UAA method with two turning points and show that the first-order approximate solution will be described by the parabolic cylinder functions. In Section 3, we construct the explicit approximate analytical solutions with the PT potential and find that the parameter space can be divided into three different cases: (A) $k^2 \gg \beta^2$, (B) $k^2 \simeq \beta^2$, and (C) $k^2 \ll \beta^2$, where $k$ and $\beta$ are real constants. After working out the error control function $\mathscr{T}$ (cf. Appendix C) in each case, we are able to determine the parameter $q_0$, which is introduced in the process of the UAA method in order to minimize the errors. Then, we show the upper bounds of errors of our approximate solutions with respect to the exact one given in Appendix B. In particular, in Case A), the upper bounds are $\lesssim 0.15\%$, while in Case B), they are no larger than 10%. In Case C), the errors are also very small, except for the minimal points (cf. Figure 10), at which the approximate solutions deviate significantly from the analytical one. The causes of such large errors are not known and are still under our investigation. In each of these three cases, we also develop our numerical codes and find that the numerical solutions trace the exact one very well and that the upper bounds of errors are always less than $10^{-4}\%$ in each of the three cases. The paper ends with Section 4, in which our main conclusions are summarized. There are also three Appendices A–C, in which some mathematical formulas are presented.

## 2. The Uniform Asymptotic Approximation Method

Let us start with the following second-order differential equation

$$\frac{d^2 \mu_k(y)}{dy^2} = f(y) \mu_k(y). \tag{7}$$

It should be noted that all second-order linear homogeneous ordinary differential equations (ODEs) can be written in the above form by properly choosing the variable $y$ and $\mu_k(y)$. Instead of working with the above form, we introduce two functions $g(y)$ and $q(y)$, so that the function $f(y)$ takes the form [62]

$$f(y) = \lambda^2 g(y) + q(y), \tag{8}$$

where $\lambda$ is a large positive dimensionless constant and serves as a bookmark, so we can expand $\mu_k(y)$ as

$$\mu_k(y) = \sum_{n=0}^{\infty} \frac{\mu_k^{(n)}(y)}{\lambda^n}. \tag{9}$$

After all the calculations are done, one can always set $\lambda = 1$ by simply absorbing the factor $\lambda^{-n}$ into $\mu_k^{(n)}(y)$. It should be noted that there exist cases in which the above expansion does not converge, and in these cases, we shall expand $\mu_k(y)$ only to finite terms, say, $\mathcal{N}$, so that $\mu_k(y)$ is well-approximated by the sum of these $\mathcal{N}$ terms. On the other hand, the main reason to introduce two functions $g(y)$ and $q(y)$, instead of only $f(y)$, is to minimize errors by properly choosing $g(y)$ and $q(y)$.

In general, the function $g(y)$ has singularities and/or zeros over the interval of our interest. We call the zeros and singularities of $g(y)$ *turning points* and *poles*, respectively. The *uniform asymptotic approximate* (UAA) solutions of $\mu_k(y)$ depend on the properties of $g(y)$ around their poles and turning points [62–64]. The cases in which $g(y)$ has both poles and turning points were studied in detail in [67,69,74], so in this paper, we shall focus ourselves on the cases where singularities are absent and only turning points exist. As to

be shown below, the treatments of these cases will be different from the ones considered in [67,69,74]. In particular, in our previous studies, the function $q(y)$ was uniquely determined by requiring that *the error control function be finite and minimized at the poles*, while in the current cases, no such poles exist. So to fix $q(y)$, other analyses of the error control function must be carried out.

### 2.1. The UAA Method

The UAA method includes three major steps: (i) the Liouville transformations, (ii) the minimization of the error control function, and (iii) the choice of the function $y(\zeta)$, where $\zeta$ is a new variable. In the following, we shall consider each of them separately.

#### 2.1.1. The Liouville Transformations

The Liouville transformations consist of introducing a new variable $\zeta(y)$, for which it is assumed that *the inverse $y = y(\zeta)$ always exists and is thrice-differentiable*. Without loss of generality, we also assume that $y(\zeta)$ is a monotonically increasing function (cf. Figure 1). Then, in terms of $U(\zeta)$, which is defined by

$$U(\zeta) \equiv \dot{y}^{-1/2} \mu_k, \tag{10}$$

Equation (7) takes the form

$$\frac{d^2 U(\zeta)}{d\zeta^2} = \left[ \lambda^2 \dot{y}^2 g + \psi(\zeta) \right] U(\zeta), \tag{11}$$

where

$$\dot{y} \equiv \frac{dy(\zeta)}{d\zeta} > 0, \quad \zeta'(y) \equiv \frac{d\zeta(y)}{dy} = \frac{1}{\dot{y}}, \tag{12}$$

and

$$\begin{aligned} \psi(\zeta) &\equiv \dot{y}^2 q + \dot{y}^{1/2} \frac{d^2}{d\zeta^2} \left( \dot{y}^{-1/2} \right) \\ &= \dot{y}^2 q - \dot{y}^{3/2} \frac{d^2}{dy^2} \left( \dot{y}^{1/2} \right) \equiv \psi(y). \end{aligned} \tag{13}$$

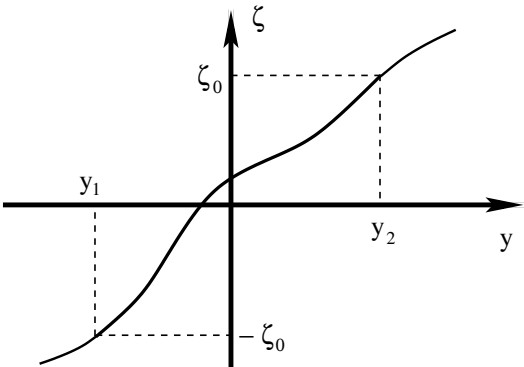

**Figure 1.** The function $\zeta(y)$ vs. $y$, which is assumed to be an always increasing function of $y$.

It should be noted that Equations (7) and (11) are completely equivalent, and so far, no approximations have been taken. However, the advantage of the form of Equation (11) is that by properly choosing $q(y)$, the term $|\psi(\zeta)|$ can be much smaller than $|\lambda^2 \dot{y}^2 g|$: that is,

$$\left| \frac{\psi}{\lambda^2 \dot{y}^2 g} \right| \ll 1, \tag{14}$$

so that the exact solution of Equation (7) can be well-approximated by the first-order solution of Equation (11) with $\psi(\zeta) = 0$. This immediately raises the question: how to choose $q(y)$ so that condition (14) holds. To explain this in detail, let us move onto the next subsection.

### 2.1.2. Minimization of Errors

To minimize the errors, let us first introduce *the error control function* [62–64,67,69,74]

$$\mathscr{T}(\zeta) \equiv -\int \frac{\psi(\zeta)}{\left|\dot{y}^2 g\right|^{1/2}} d\zeta. \tag{15}$$

Then, introducing the free parameters $a_n$ and $b_n$ into the functions $g(y)$ and $q(y)$, we have

$$g(y) = g(y, a_n), \quad q(y) = q(y, b_n), \tag{16}$$

where $n = 1, 2, ..., N$, with $N$ being an integer. It is clear that for such chosen $g(y)$ and $q(y)$, the error control function $\mathscr{T}(\zeta)$ will also depend on $a_n$ and $b_n$. To minimize the errors, one way is to minimize the error control function by properly choosing $a_n$ and $b_n$ so that

$$\frac{\partial \mathscr{T}(\zeta, a_n, b_n)}{\partial a_n} = 0, \qquad \frac{\partial \mathscr{T}(\zeta, a_n, b_n)}{\partial b_n} = 0,$$
$$(n = 1, 2, ..., N). \tag{17}$$

### 2.1.3. Choice of $y(\zeta)$

On the other hand, the errors also depend on the choice of $y(\zeta)$, which in turn sensitively depends on the properties of the functions $g(y)$ and $q(y)$ near their poles and turning points. In addition, it must be chosen so that the resulting equation of the first-order approximation (obtained by setting $\psi(\zeta) = 0$) can be solved explicitly (in terms of known functions). Considering all the above, it has been found that $y(\zeta)$ can be chosen as [62–64,67,69,74]

$$\dot{y}^2 g = \begin{cases} \text{sgn}(g), & \text{zero turning point,} \\ \zeta, & \text{one turning point,} \\ \zeta_0^2 - \zeta^2, & \text{two turning point,} \end{cases} \tag{18}$$

in the cases with zero, one, and two turning points, respectively. Here $\text{sgn}(g) = 1$ for $g > 0$, and $\text{sgn}(g) = -1$ for $g < 0$.

In the rest of this paper, we shall consider only the cases with two turning points.

### 2.2. UAA Method for Two Turning Points

For the cases with two turning points, we can always write $g(y)$ as

$$g(y) = p(y)(y - y_1)(y - y_2), \tag{19}$$

where $y_1$ and $y_2$ are the two turning points, and $p(y)$ is a function of $y$, with $p(y_i) \neq 0$, $(i = 1, 2)$. In general, according to the properties of $y_1$ and $y_2$, we can divide all the cases into three different subclasses:

1.　$y_1$ and $y_2$ are two distinct real roots of $g(y) = 0$;
2.　$y_1 = y_2$, a double real root of $g(y) = 0$;
3.　$y_1$ and $y_2$ are two complex roots of $g(y) = 0$. Since $g(y)$ is real, in this case, these two roots must be complex conjugate, $y_1 = y_2^*$.

To apply the UAA method to Equation (12), we assume that the following conditions are satisfied [67,69,74]:

- When far away from any of the two turning points, we require

$$\left| \frac{q(y)}{g(y)} \right| \ll 1. \tag{20}$$

- When near any of these two points, we require

$$\left| \frac{q(y)(y - y_i)}{g(y)} \right| \ll 1, \ (i = 1, 2), \tag{21}$$

provided that the two turning points are far away from each other: that is, when $|y_1 - y_2| \gg 1$.

- If the two turning points are close to each other, $|y_1 - y_2| \simeq 0$, then near these points, we require

$$\left| \frac{q(y)(y - y_1)(y - y_2)}{g(y)} \right| \ll 1. \tag{22}$$

It should be noted that when $|y_2 - y_1| \gg 1$, the two turning points are far away, and each of them can be treated as an isolated single turning point [62,63]. In addition, without loss of generality, we assume that $g(y) < 0$ for $y > y_2$ or $y < y_1$ when $y_1$ and $y_2$ are real. When $y_2$ and $y_1$ are complex conjugate, we assume that $g(y) < 0$ (cf. Figure 2). Then, in this case, we adopt a method to treat all these three classes listed above together [64,67,69,74]. In particular, we choose $\dot{y}^2 g$ as

$$\dot{y}^2 g = \zeta_0^2 - \zeta^2 \begin{cases} > 0, & g > 0, \\ = 0, & g = 0, \\ < 0, & g < 0, \end{cases} \tag{23}$$

so that $\zeta$ is an increasing function of $y$ (cf. Figure 1) and

$$\sqrt{|g(y)|}\, dy = \sqrt{|\zeta_0^2 - \zeta^2|}\, d\zeta. \tag{24}$$

When we integrate the above equation, without loss of generality, we shall choose the integration constants so that

$$\zeta(y_1) = -\zeta_0, \quad \zeta(y_2) = \zeta_0. \tag{25}$$

Then, we find that

$$\zeta_0^2 = \begin{cases} > 0, & y_{1,2} \text{ real, and } y_1 \neq y_2, \\ = 0, & y_{1,2} \text{ real, and } y_1 = y_2, \\ < 0, & y_{1,2} \text{ complex,} \end{cases} \tag{26}$$

with

$$\begin{aligned} \zeta_0^2 &= \pm \frac{2}{\pi} \int_{y_1}^{y_2} \sqrt{|g(y)|}\, dy \\ &= \pm \frac{2}{\pi} \int_{-\zeta_0}^{\zeta_0} \sqrt{|\zeta_0^2 - \zeta^2|}\, d\zeta, \end{aligned} \tag{27}$$

where " $+$ " corresponds to cases in which the two turning points $y_1$ and $y_2$ are both real, and " $-$ " corresponds to cases in which the two turning points $y_1$ and $y_2$ are complex conjugate. When $y_1$ and $y_2$ are complex conjugate, the integration of Equation (27) is along the imaginary axis [64]. When the two real roots are equal, we have $\zeta_0 = 0$.

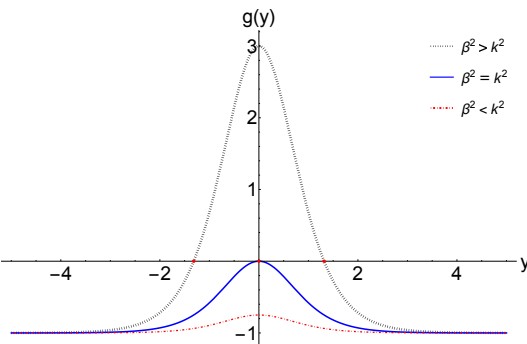

**Figure 2.** The function $g(y)$ defined by Equation (39) for different choices of $k$ and $\beta$. In particular, the dotted black line denotes the case $k^2 < \beta^2$, and the solid blue line denotes the case $k^2 = \beta^2$, while the dash-dotted red line denotes the case $k^2 > \beta^2$.

To proceed further, let us first derive the relation between $\zeta(y)$ and $y$ by first integrating the right-hand side of Equation (24). To this goal, it is found to be easier to distinguish the case in which $y_1$ and $y_2$ are real from the one in which they are complex conjugate.

2.2.1. When $y_{1,2}$ Are Real

Let us first consider the case when $y_1$ and $y_2$ are real. Then, when $y > y_2$, we have $\zeta(y) > \zeta_0$ (cf. Figure 1). Hence, from Equation (24), we find

$$
\int_{y_2}^{y} \sqrt{-g(y')}dy' = \int_{\zeta_0}^{\zeta} \sqrt{v^2 - \zeta_0^2}dv
$$

$$
= \frac{1}{2}\zeta\sqrt{\zeta^2 - \zeta_0^2} - \frac{\zeta_0^2}{2}\ln\left(\frac{\zeta + \sqrt{\zeta^2 - \zeta_0^2}}{\zeta_0}\right)
$$

$$
= \frac{1}{2}\zeta\sqrt{\zeta^2 - \zeta_0^2} - \frac{\zeta_0^2}{2}\operatorname{arcosh}\left(\frac{\zeta}{\zeta_0}\right), \quad (y \geq y_2). \tag{28}
$$

When $y \leq y_1$, we have $\zeta(y) \leq -\zeta_0$. Then, from Equation (24), we find

$$
\int_{y}^{y_1} \sqrt{-g(y')}dy' = \int_{\zeta}^{-\zeta_0} \sqrt{v^2 - \zeta_0^2}dv
$$

$$
= -\frac{1}{2}\zeta\sqrt{\zeta^2 - \zeta_0^2} + \frac{\zeta_0^2}{2}\ln\left(\frac{-\zeta - \sqrt{\zeta^2 - \zeta_0^2}}{\zeta_0}\right)
$$

$$
= -\frac{1}{2}\zeta\sqrt{\zeta^2 - \zeta_0^2} - \frac{\zeta_0^2}{2}\ln\left(\frac{-\zeta + \sqrt{\zeta^2 - \zeta_0^2}}{\zeta_0}\right)
$$

$$
= -\frac{1}{2}\zeta\sqrt{\zeta^2 - \zeta_0^2} - \frac{\zeta_0^2}{2}\operatorname{arcosh}\left(-\frac{\zeta}{\zeta_0}\right), \quad (y \leq y_1). \tag{29}
$$

When $y_1 \leq y \leq y_2$, we have $-\zeta_0 < \zeta(y) < \zeta_0$, and

$$
\int_{y_1}^{y} \sqrt{g(y')}dy' = \int_{-\zeta_0}^{\zeta} \sqrt{\zeta_0^2 - v^2}dv = \frac{1}{2}\zeta\sqrt{\zeta_0^2 - \zeta^2}
$$

$$
+ \frac{\zeta_0^2}{2}\arccos\left(-\frac{\zeta}{\zeta_0}\right), \quad (y_1 \leq y \leq y_2). \tag{30}
$$

### 2.2.2. When $y_{1,2}$ Are Complex Conjugate

Now let us turn to consider the case when $y_1$ and $y_2$ are complex. For this case, $\zeta_0^2$ is always negative, $\zeta_0^2 < 0$; thus, from Equation (10), we find [64]

$$
\begin{aligned}
\int_0^y \sqrt{-g(y')}dy' &= \int_0^\zeta \sqrt{\zeta^2 - \zeta_0^2}d\zeta \\
&= \frac{1}{2}\zeta\sqrt{\zeta^2 - \zeta_0^2} - \frac{\zeta_0^2}{2}\ln\left(\frac{\zeta + \sqrt{\zeta^2 - \zeta_0^2}}{|\zeta_0|}\right).
\end{aligned}
\tag{31}
$$

### 2.2.3. The First-Order Approximate Solutions

With the choice of Equation (23), we find that Equation (12) reduces to

$$
\frac{d^2 U}{d\zeta^2} = \left[\lambda^2\left(\zeta_0^2 - \zeta^2\right) + \psi(\zeta)\right]U,
\tag{32}
$$

where we assume that $\zeta \in (-\zeta_2, \zeta_2)$, with $\zeta_2$ being a real and positive constant, which can be arbitrarily large $\zeta_2 \to \infty$.

Neglecting the $\psi(\zeta)$ term, we find that the approximate solutions can be expressed in terms of the parabolic cylinder functions $W(\frac{1}{2}\lambda\zeta_0^2, \pm\sqrt{2\lambda}\zeta)$ [64], and are given by

$$
\begin{aligned}
U(\zeta) &= \alpha_k\left\{W\left(\frac{1}{2}\lambda\zeta_0^2, \sqrt{2\lambda}\zeta\right) + \epsilon_1\right\} \\
&\quad +\beta_k\left\{W\left(\frac{1}{2}\lambda\zeta_0^2, -\sqrt{2\lambda}\zeta\right) + \epsilon_2\right\},
\end{aligned}
\tag{33}
$$

from which we have

$$
\begin{aligned}
\mu_k(y) &= \alpha_k\left(\frac{\zeta^2 - \zeta_0^2}{-g(y)}\right)^{\frac{1}{4}}\left[W\left(\frac{1}{2}\lambda\zeta_0^2, \sqrt{2\lambda}\zeta\right) + \epsilon_1\right] \\
&\quad +\beta_k\left(\frac{\zeta^2 - \zeta_0^2}{-g(y)}\right)^{\frac{1}{4}}\left[W\left(\frac{1}{2}\lambda\zeta_0^2, -\sqrt{2\lambda}\zeta\right) + \epsilon_2\right],
\end{aligned}
\tag{34}
$$

where $\alpha_k$ and $\beta_k$ are two integration constants, and $\epsilon_1$ and $\epsilon_2$ are the errors of the corresponding approximate solutions, whose upper bounds are given by Equations (A1) and (A2) in Appendix A.

For the choice of Equation (23), we find that the associated error control function defined by Equation (15) now takes the form

$$
\begin{aligned}
\mathcal{T}(\zeta) &= -\int^\zeta\left\{\frac{q}{g} - \frac{5}{16}\frac{g'^2}{g^3} + \frac{1}{4}\frac{g''}{g^2}\right\}\sqrt{v^2 - \zeta_0^2}dv \\
&\quad +\int^\zeta\left\{\frac{5\zeta_0^2}{4(v^2 - \zeta_0^2)^3} + \frac{3}{4(v^2 - \zeta_0^2)^2}\right\}\sqrt{v^2 - \zeta_0^2}dv \\
&= -\int^y\left\{\frac{q}{g} - \frac{5}{16}\frac{g'^2}{g^3} + \frac{1}{4}\frac{g''}{g^2}\right\}\sqrt{-g}dy' \\
&\quad +\int^\zeta\left\{\frac{5\zeta_0^2}{4(v^2 - \zeta_0^2)^{5/2}} + \frac{3}{4(v^2 - \zeta_0^2)^{3/2}}\right\}dv,
\end{aligned}
\tag{35}
$$

for $g < 0$, and

$$
\begin{aligned}
\mathscr{T}(\zeta) &= \int^{\zeta} \left\{ \frac{q}{g} - \frac{5}{16}\frac{g'^2}{g^3} + \frac{1}{4}\frac{g''}{g^2} \right\} \sqrt{\zeta_0^2 - v^2}\, dv \\
&\quad - \int^{\zeta} \left\{ \frac{5\zeta_0^2}{4(v^2 - \zeta_0^2)^3} + \frac{3}{4(v^2 - \zeta_0^2)^2} \right\} \sqrt{\zeta_0^2 - v^2}\, dv \\
&= \int^{y} \left\{ \frac{q}{g} - \frac{5}{16}\frac{g'^2}{g^3} + \frac{1}{4}\frac{g''}{g^2} \right\} \sqrt{g}\, dy' \\
&\quad + \int^{\zeta} \left\{ \frac{5\zeta_0^2}{4(\zeta_0^2 - v^2)^{5/2}} - \frac{3}{4(\zeta_0^2 - v^2)^{3/2}} \right\} dv,
\end{aligned}
\tag{36}
$$

for $g > 0$.

### 3. UAA Solutions with the Pöschl–Teller Potential

To study the case in which only turning points exist, in this paper, we consider the second-order differential Equation (7) with a Pöschl–Teller (PT) potential [87,88]

$$
\left( \lambda^2 g + q \right) = - \left( k^2 - \frac{\beta_0^2}{\cosh^2(\alpha y)} \right),
\tag{37}
$$

as in this case exact solutions exist, where $k$ is the comoving wavenumber, and $\beta_0$ is a real and positive constant. The two parameters $\beta_0$ and $\alpha$ determine the height and the spread of the PT potential, respectively. Under the rescaling $\alpha y \to y$, the $\alpha$ parameter can be absorbed into the wavenumber $k$ and $\beta_0$ by redefining $(k/\alpha \to k, \beta_0/\alpha \to \beta_0)$. As a result, there is no loss of generality to set $\alpha = 1$ from now on. Then, the exact solutions in this case exist and are presented in Appendix B.

On the other hand, to apply the UAA method to this case and to minimize the errors of the analytic approximate solutions, we tentatively choose $q$ as

$$
q = \frac{q_0^2}{\cosh^2(y)},
\tag{38}
$$

where $q_0$ is a free parameter to be determined below by minimizing the error control function (15) with the choice of $\dot{y}^2 g$ given by Equation (23). Then, we have

$$
g(y) = \frac{\beta^2}{\cosh^2(y)} - k^2,
\tag{39}
$$

where $\beta \equiv \sqrt{\beta_0^2 - q_0^2}$. In this paper, without loss of generality, we shall choose $q_0$ so that $\beta$ is always real: that is

$$
\beta^2 \equiv \beta_0^2 - q_0^2 > 0.
\tag{40}
$$

Thus, from $g(y) = 0$ we find that the two roots are given by

$$
y_i = \pm \cosh^{-1} \frac{\beta}{k} = \pm \cosh^{-1} \frac{\sqrt{\beta_0^2 - q_0^2}}{k}.
\tag{41}
$$

It is clear that, depending on the relative magnitudes of $\beta_0$ and $k$ as well as the choices of $q_0$, two turning points can be either complex or real. In Figure 2, we plot out the three different cases, $k^2 < \beta^2$, $k^2 = \beta^2$, and $k^2 > \beta^2$, from which it can be seen clearly that the two turning

points are real and different for $k^2 < \beta^2$, real and equal for $k^2 = \beta^2$, and complex conjugate for $k^2 > \beta^2$. Then, from Equations (38) and (39), we find that

$$\left| \frac{q(y)}{g(y)} \right| = \left| \frac{q_0^2}{\beta^2 - k^2 \cosh^2(y)} \right| \simeq q_0^2 e^{-2|y|} \tag{42}$$

for $|y| \gg 0$,

$$\left| \frac{q(y)(y - y_i)}{g(y)} \right| \simeq \frac{q_0^2}{y + y_j}, (i \neq j) \tag{43}$$

for $|y| \simeq |y_i|$ and $|y_1 - y_2| \gg 1$, and

$$\left| \frac{q(y)(y - y_1)(y - y_2)}{g(y)} \right| \simeq q_0^2 \tag{44}$$

for $|y| \simeq |y_1|$ and $|y_1 - y_2| \simeq 0$. In the following, let us consider the three cases: (a) $k^2 \gg \beta^2$, (b) $k^2 \simeq \beta^2$, and (c) $\beta^2 \gg k^2$ separately.

*3.1. $k^2 \gg \beta^2$*

In this case, we have $g(y)$ is always negative, $g(y) < 0$, so that the two turning points of $g(y) = 0$ are complex conjugate and are given by

$$y_1 = y_2^* = -i \cos^{-1}\left( \frac{\beta}{k} \right) \simeq -\frac{i\pi}{2}. \tag{45}$$

As discussed in the last section, now $\zeta_0^2 < 0$, for which Equation (32) can be cast in the form

$$\frac{d^2 W(\zeta)}{d^2 \zeta} = \left\{ -\lambda^2 \left( \zeta^2 + \hat{\zeta}_0^2 \right) + \psi \right\} W(\zeta), \tag{46}$$

where $\hat{\zeta}_0^2 \equiv -\zeta_0^2 > 0$. Note that in writing down the above equation, we replaced $U$ with $W$. In addition, the new variable $\zeta$ is related to $y$ via

$$\int_0^y \sqrt{-g(y)} dy = \int_0^\zeta \sqrt{v^2 + \hat{\zeta}_0^2} dv = \frac{1}{2} \hat{\zeta}_0^2 \ln\left( \zeta + \sqrt{\zeta^2 + \hat{\zeta}_0^2} \right)$$
$$+ \frac{1}{2} \zeta \sqrt{\zeta^2 + \hat{\zeta}_0^2} - \frac{1}{2} \hat{\zeta}_0^2 \ln \hat{\zeta}_0, \tag{47}$$

from which we find that $\hat{\zeta}_0$ is given explicitly by

$$\hat{\zeta}_0^2 = 2(k - \beta) > 0. \tag{48}$$

Moreover, in the case of the PT potential, the integration of Equation (47) can be carried out explicitly, giving

$$\int_0^y dy \sqrt{-g} = \epsilon_y \sqrt{1 - x^2} \sqrt{k^2 - \beta^2} \times \mathrm{AppellF_1}\left( \frac{1}{2}, -\frac{1}{2}, 1, \frac{3}{2}; \frac{1 - x^2}{1 - k^2/\beta^2}, 1 - x^2 \right), \tag{49}$$

where $\epsilon_y$ denotes the sign of $y$, with $x \equiv 1/\cosh(y)$, and $\mathrm{AppellF_1}$ is the Appell hypergeometric function. Ignoring the $\psi$ term in Equation (46), we find the general solution

$$\mu_k(y) = \left( \frac{\zeta^2 + \hat{\zeta}_0^2}{-g(y)} \right)^{1/4} \left\{ \alpha_k W\left( -\frac{\hat{\zeta}_0^2}{2}, \sqrt{2}\zeta \right) + \beta_k W\left( -\frac{\hat{\zeta}_0^2}{2}, -\sqrt{2}\zeta \right) \right\}, \tag{50}$$

where $W$ denotes the Weber parabolic cylinder function [89], and $\alpha_k$ and $\beta_k$ are two integration parameters that generally depend on the comoving wavenumber $k$.

The validity of the analytic solution (50) depends on the criteria given by Equations (20)–(22), while its accuracy can be predicted by the error control function $\mathscr{T}$. In the current case, we find that $\mathscr{T}$ of Equation (35) can be written as a combination of the three terms given by Equations (A10) [2], where

$$
\begin{aligned}
\mathscr{T}_1 &= \int_0^y \frac{q}{\sqrt{-g}} dy = \frac{q_0^2 \epsilon_y}{\beta} \ln\left( \frac{\sqrt{1-x^2}\beta + \sqrt{k^2 - \beta^2 x^2}}{\sqrt{k^2 - \beta^2}} \right), \\
\mathscr{T}_2 &= \int_0^y \left( \frac{5g'^2}{16g^3} - \frac{g''}{4g^2} \right) \sqrt{-g}\, dy = -\epsilon_y \left\{ \frac{1}{4\beta} \ln\left( \frac{\sqrt{1-x^2}\beta + \sqrt{k^2 - x^2\beta^2}}{\sqrt{k^2 - \beta^2}} \right) \right. \\
&\qquad\qquad \left. - \frac{\sqrt{1-x^2}A}{12(k^2 - \beta^2)(k^2 - \beta^2 x^2)^{3/2}} \right\}, \\
\mathscr{T}_3 &= \int_0^\zeta \left( \frac{-5\hat{\zeta}_0^2}{4\left(v^2 + \hat{\zeta}_0^2\right)^{5/2}} + \frac{3}{4\left(v^2 + \hat{\zeta}_0^2\right)^{3/2}} \right) dv = -\frac{\zeta\left(\zeta^2 + 6\hat{\zeta}_0^2\right)}{12\hat{\zeta}_0^2\left(\zeta^2 + \hat{\zeta}_0^2\right)^{3/2}},
\end{aligned}
\tag{51}
$$

where $A$ is given by Equation (A12). It should be noted that $\mathscr{T}_1$, $\mathscr{T}_2$, and $\mathscr{T}_3$ given in Equation (51) all vanish when $y = 0$ (for which we have $x = 1$ and $\zeta = 0$); that is,

$$
\mathscr{T}(\zeta = 0) = 0.
\tag{52}
$$

Further, as the PT potential is an even function, the error control function is antisymmetric about the origin: namely, $\mathscr{T}(-y) = -\mathscr{T}(y)$. As a result, we will study its behavior only on the positive $y$ axis, $y \geq 0$. With the help of Equation (47), the numeric value of the error control function at any point $y > 0$ can be found from Equation (51). In particular, for $\beta/k \ll 1$, we find that

$$
\begin{aligned}
\mathscr{T} &= \frac{q_0^2}{k}\sqrt{1 - x^2} - \frac{\zeta\left(\zeta^2 + 6\hat{\zeta}_0^2\right)}{12\hat{\zeta}_0^2\left(\zeta^2 + \hat{\zeta}_0^2\right)^{3/2}} + \mathcal{O}\left(x^2, \frac{\beta^2}{k^3}\right) \\
&\to \frac{1}{24k}\left[ \left(24q_0^2 - 1\right) - \left(\frac{\beta}{k}\right) + \mathcal{O}\left(\frac{\beta^2}{k^2}\right) \right],
\end{aligned}
\tag{53}
$$

as $x \to 0$ (or $y \to \infty$). Note that $\zeta \to \infty$ as $y \to \infty$, which can be seen clearly from Equation (47). Thus, to minimize the error control function for very large values of $y$, we must choose

$$
q_0^2 = \frac{1}{24} \simeq 4.167 \times 10^{-2}.
\tag{54}
$$

In Figure 3, we plot the functions $|q/g|$, $|q(y - y_1)/g|$, and $|q(y - y_1)(y - y_2)/g|$ together with the error control function defined by Equations (A10)–(A12) for $(k, \beta) = (5.0, 1.0)$, with $q_0$ being given by Equation (54). (Recall $\beta_0 \equiv \sqrt{\beta^2 + q_0^2}$.) From these figures, it is clear that the conditions (20)–(22) are well-satisfied, and the error control function remains small all the time. In particular, it decreases as $\beta/k$ decreases.

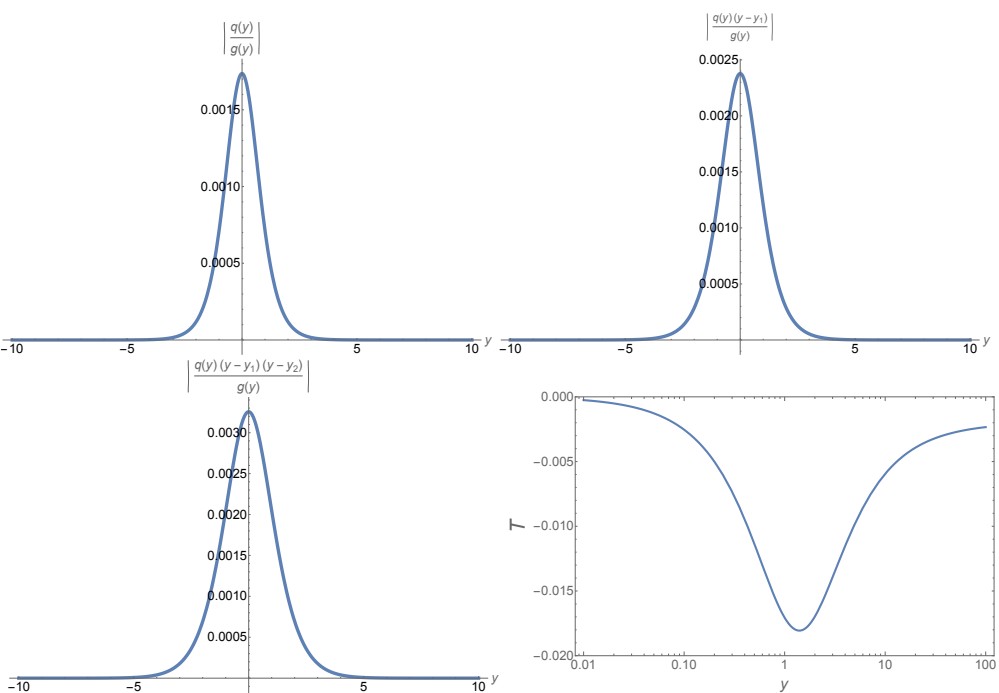

**Figure 3.** Plots of the quantities $|q/g|$, $|q(y - y_1)/g|$, $|q(y - y_1)(y - y_2)/g|$, and the error control function $\mathcal{T}$ for $k = 5.0$, $\beta = 1.0$, and $q_0 = 1/\sqrt{24}$, for which we have $y_2 = y_1^* = 1.36944i$.

In Figure 4, we plot the mode functions $\mu_k(y)$, $\mu_k^N(y)$, $\mu_k^E(y)$, and the relative difference $\delta^A(y)$ defined by

$$\delta^A(y) \equiv \left| \frac{|\mu_k(y)| - |\mu_k^A(y)|}{\mu_k^A(y)} \right|, \tag{55}$$

where $A = (N, E)$, $\mu_k(y)$ denotes the mode function obtained by the UAA method given by Equation (50), $\mu_k^N(y)$ is the numerical solution obtained by integrating Equation (7) directly with the same initial conditions, while $\mu_k^E(y)$ is the exact solution given by Equation (A7). From these figures, we can see that the maximal errors occur in the region near $y = 0$, but the upper bound is no larger than 0.15% at any given $y$, including the region near $y \simeq 0$.

It is interesting to note that this analytical approximate solution is only up to the first-order approximation of the UAA method. With higher-order approximations, the relative errors are even smaller.

To check our numerical solutions, in Figure 4 we also plot the relative differences $\epsilon(y)$ between $\mu_k^N(y)$ and $\mu_k^E(y)$, defined by

$$\epsilon(y) \equiv \left| \frac{|\mu_k^N(y)| - |\mu_k^E(y)|}{\mu_k^E(y)} \right|. \tag{56}$$

From these figures, it can be seen that $\epsilon(y)$ is no larger than $10^{-7}$, and our numerical code is well-tested and justified.

It is also interesting to note that the mode functions are oscillating for $y \lesssim -10$, and these fine features are captured in all three mode functions, although there are some differences in the details. Again, as shown by their relative variations, these differences are very small. In addition, we also consider other choices of $\beta$ and $k$ and find that they all have similar properties under the condition $k^2 \gg \beta^2$.

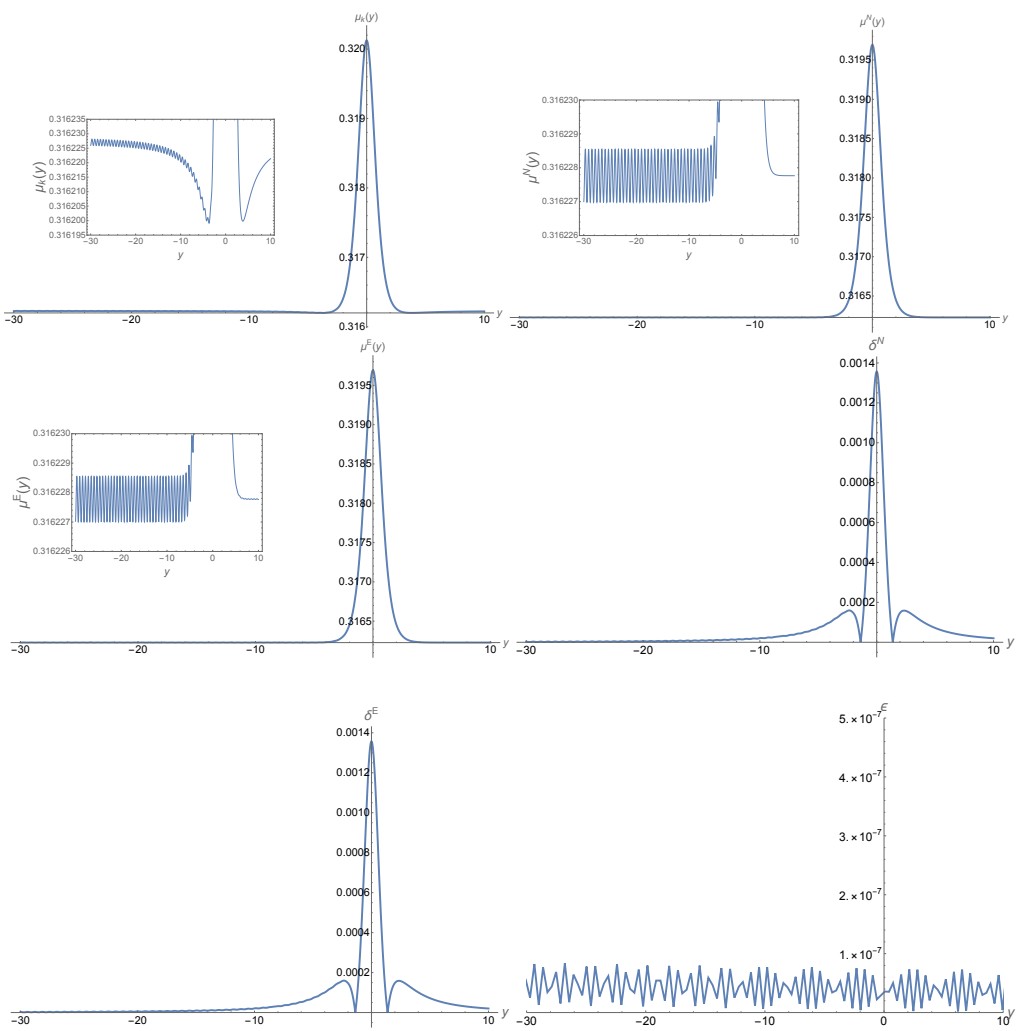

**Figure 4.** Plots of the mode functions $\mu_k(y)$, $\mu_k^N(y)$, $\mu_k^E(y)$ and their relative differences $\delta^N(y)$, $\delta^E(y)$ and $\epsilon(y)$ for $k = 5.0$, $\beta = 1.0$ and $q_0 = 1/\sqrt{24}$, for which we have $y_2 = y_1^* = 1.36944i$.

### 3.2. $\beta^2 \simeq k^2$

In this case, depending on $k \gtrsim \beta$ or $k \lesssim \beta$, the function $g(y)$ has different properties, as shown in Figure 2. Therefore, in the following subsections, let us consider them separately.

### 3.2.1. $k \gtrsim \beta$

When $k \gtrsim \beta$, the function $g(y)$ is always non-positive for $y \in (-\infty, \infty)$. Then from Equations (A10) and (51), we find that

$$\mathcal{T}(y) \simeq \frac{q_0^2 - 1/4}{2\beta} \ln\left(\frac{2}{\epsilon}\right) + \frac{9}{48k} + \mathcal{O}(\epsilon),$$

(57)

as $y \to \infty$, but now we have $\epsilon \equiv (k - \beta)/k$. Thus, to have the error control function be finite at $y = \infty$, now we must set

$$q_0^2 = \frac{1}{4},$$ (58)

instead of the value given by Equation (54) for the case $k \gg \beta^2$. In Figure 5, we plot the quantities $|q/g|$, $|q(y - y_1)/g|$, $|q(y - y_1)(y - y_2)/g|$, and the error control function $\mathcal{T}$ for $k = 5.0$, $\beta = 4.9$, and $q_0 = 1/2$, for which we have $y_1 = y_2^* = 0.200335i$. From these figures, we can see clearly that the conditions (20)–(22) are well-satisfied, and the error control function remains small all the time. Then, the corresponding quantities $\mu_k(y)$, $\mu_k^N(y)$, $\mu_k^E(y)$, $\delta^A(y)$, and $\epsilon(y)$ are plotted in Figure 6. From the curves of $\delta^N(y)$ and $\delta^E(y)$, we can see that now the errors of the first-order UAA solution are $\leq 4\%$, which are larger than those of the last subcase. This is mainly because of the fast oscillations of the solution in the region $y < 0$. Therefore, in order to obtain solutions with high precision, high-order approximations for this case are needed. However, we do like to note that our numerical solution still matches the exact one very well, as shown by the curve of $\epsilon(y)$, which is no larger than $6.0 \times 10^{-6}$.

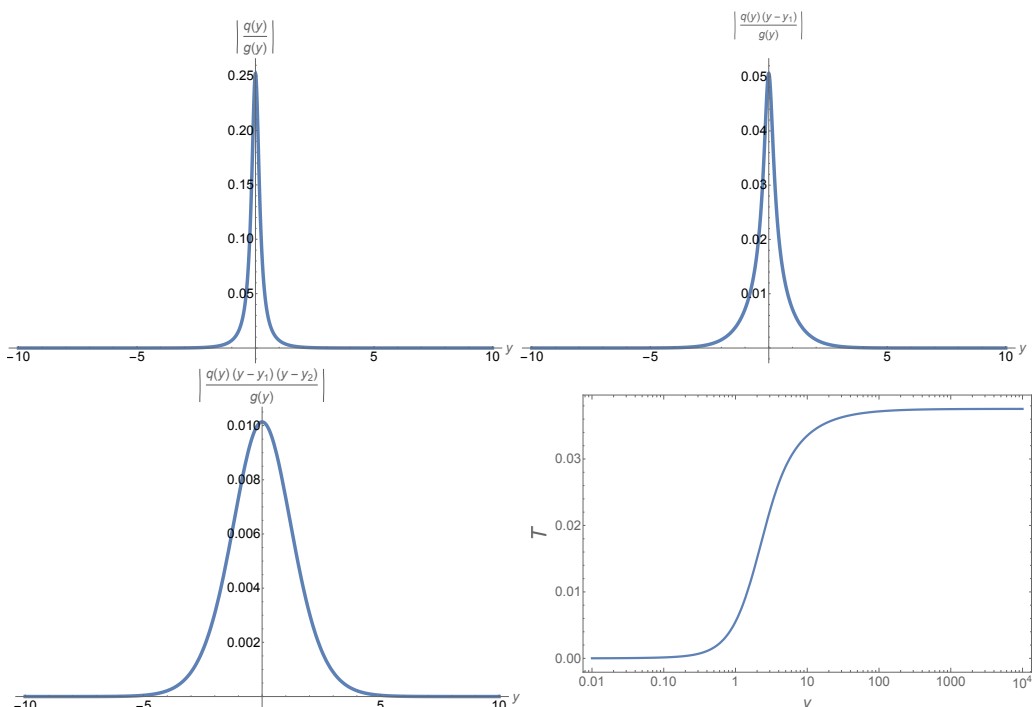

**Figure 5.** Plots of the quantities $|q/g|$, $|q(y - y_1)/g|$, $|q(y - y_1)(y - y_2)/g|$, and the error control function $\mathcal{T}$ for $k = 5.0$, $\beta = 4.9$, and $q_0 = 1/2$, for which we have $y_1 = y_2^* = 0.200335i$.

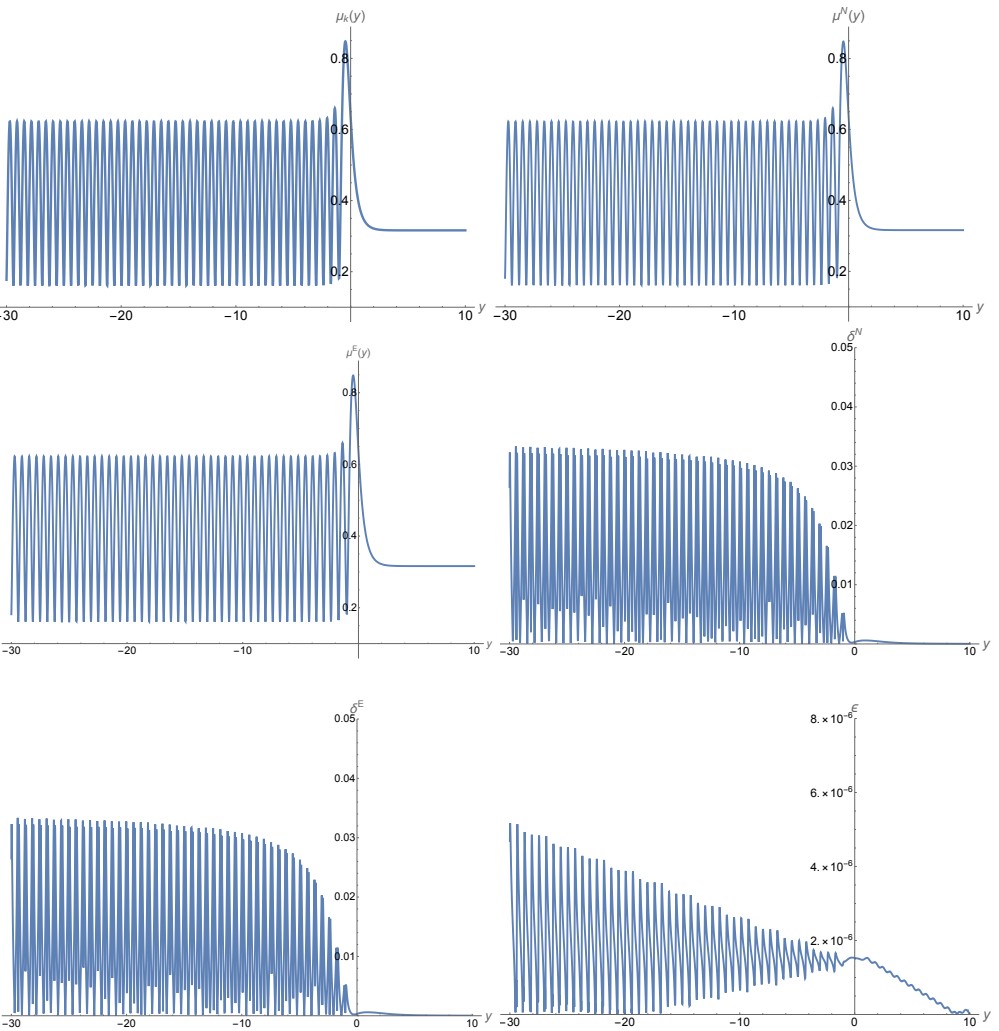

**Figure 6.** Plots of the mode functions $\mu_k(y)$, $\mu_k^N(y)$, and $\mu_k^E(y)$ and their relative differences $\delta^N(y)$, $\delta^E(y)$, and $\epsilon(y)$ for $k = 5$, $\beta = 4.9$, and $q_0 = 1/2$.

3.2.2. $k \lesssim \beta$

In this case, we find that

$$\zeta_0^2 = \frac{2}{\pi} \left| \int_{y_1}^{y_2} \sqrt{g(y)}\, dy \right| = 2|k - \beta|. \tag{59}$$

On the other hand, from Equations (36), (A10) and (A13), we find that

$$\mathscr{T}(y) \simeq \begin{cases} \dfrac{\zeta(0)\left(6\zeta_0^2 - \zeta^2(0)\right)}{12\zeta_0^2\left(\zeta_0^2 - \zeta^2(0)\right)^{3/2}}, & y \to 0, \\[4mm] \dfrac{\pi\left(q_0^2 - 1/4\right)}{2\beta}, & y \to y_2, \end{cases} \tag{60}$$

where $\zeta(0) \equiv \zeta(y)|_{y=0} < \zeta_0$. Note that in calculating the error control function near the turning point $y \simeq y_2$, we used the relation

$$\frac{\beta}{k^2\sqrt{\beta^2 - k^2}}\left(\beta^2 x^2 - k^2\right)^{3/2} \simeq \frac{1}{\zeta_0}(\zeta_0^2 - \zeta^2)^{3/2}, \tag{61}$$

so that the divergence of the second term of $\mathcal{T}_2$ cancels exactly with that of $\mathcal{T}_3$. Equation (61) can be obtained directly from the relation $\sqrt{g}dy = \sqrt{\zeta_0^2 - \zeta^2}d\zeta$ for the case $g \geq 0$. Similarly, it can be shown that

$$\mathcal{T}(y) \simeq \frac{q_0^2 - 1/4}{2\beta} \ln\left(\frac{2}{\epsilon}\right), \quad y \to \infty. \tag{62}$$

It is clear that to minimize the errors, in the present case, $q_0^2$ must also be chosen to be

$$q_0^2 = \frac{1}{4}, \tag{63}$$

as given by Equation (58). In Figure 7, we plot the quantities $|q/g|$, $|q(y - y_1)/g|$, $|q(y - y_1)(y - y_2)/g|$, and the error control function $\mathcal{T}$ for $k = 5.0$, $\beta = 5.1$, and $q_0 = 1/2$, for which we have $y_1 = -y_2 = -0.199668$. It is clear that in this case, the two turning points are very close, and the conditions $|q/g| \ll 1$ and $|q(y - y_1)/g| \ll 1$ are violated near these points. However, the condition $|q(y - y_1)(y - y_2)/g| \ll 1$ holds near them. So, conditions (20)–(22) are also satisfied, and the error control function remains small all the time.

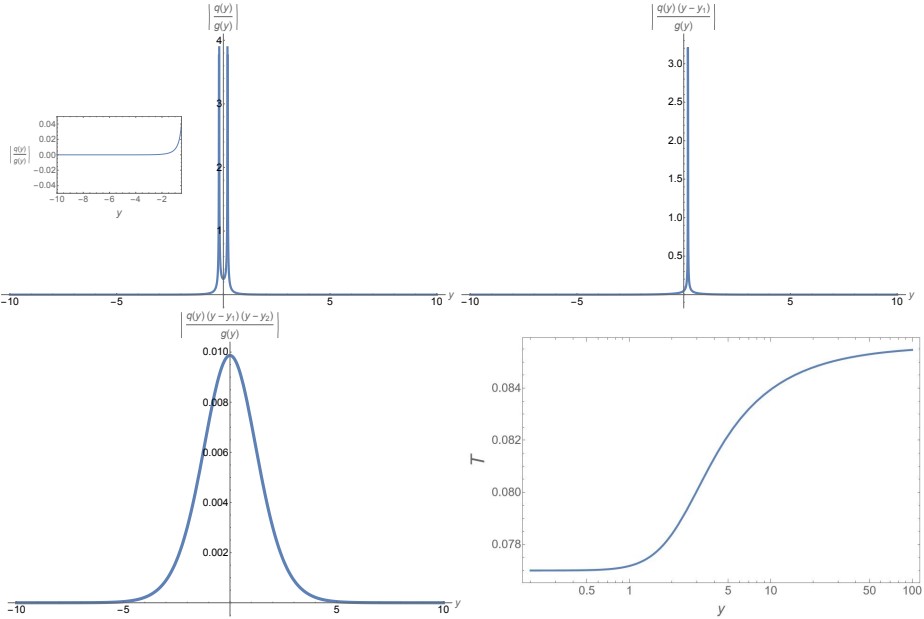

**Figure 7.** Plots of the quantities $|q/g|$, $|q(y - y_1)/g|$, $|q(y - y_1)(y - y_2)/g|$, and the error control function $\mathcal{T}$ for $k = 5.0$, $\beta = 5.1$, and $q_0 = 1/2$, for which we have $y_1 = -y_2 = -0.199668$.

Then, the corresponding quantities $\mu_k(y)$, $\mu_k^N(y)$, $\mu_k^E(y)$, $\delta^A(y)$, and $\epsilon(y)$ are plotted in Figure 8. From the curves of $\delta^N(y)$ and $\delta^E(y)$, we can see that now the errors of the first-order UAA solution are $\lesssim 10\%$. Similar to the last subcase, this is mainly because of the fast oscillations of the solution in the region $y < 0$. Therefore, in order to obtain high precision, high-order approximations for this case are needed, too. In addition, our numerical solution still matches well with the exact one, as shown by the curve of $\epsilon(y)$, which is no larger than $2.0 \times 10^{-6}$.

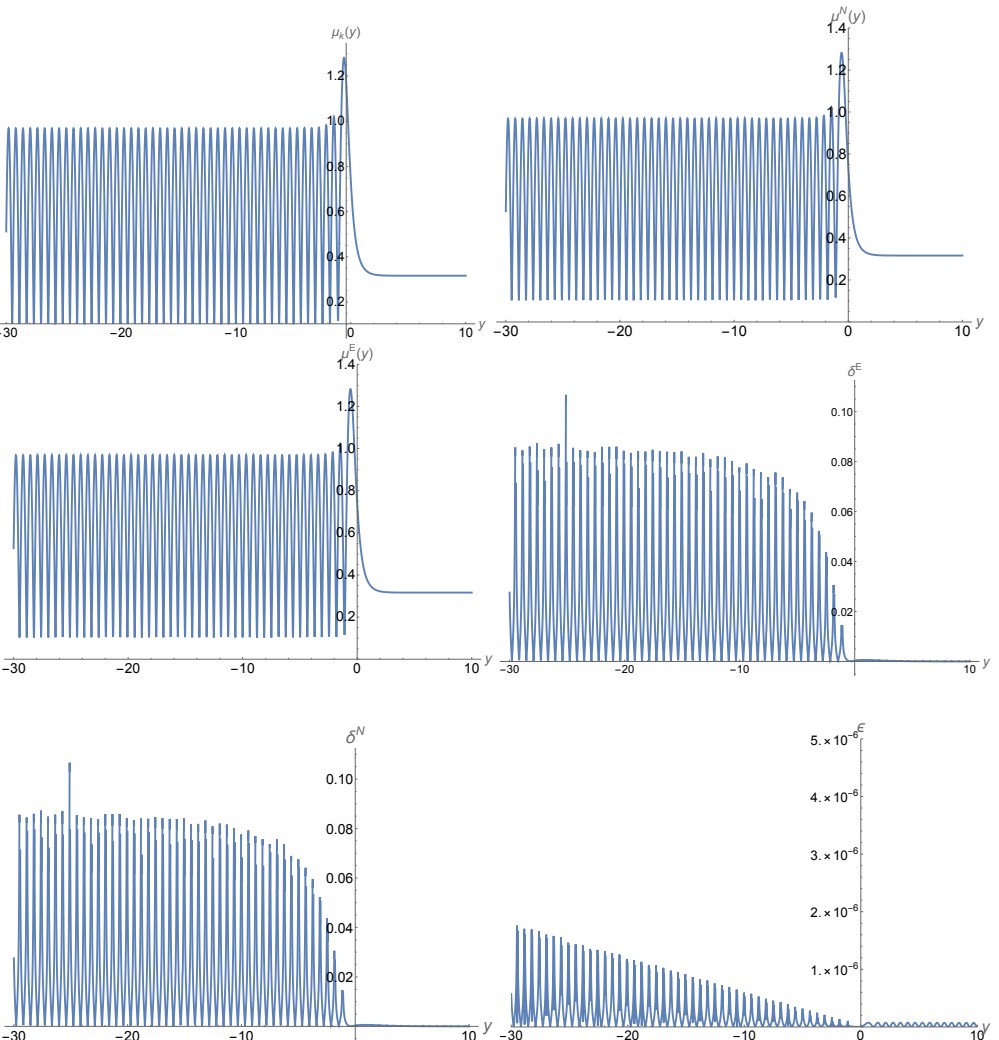

**Figure 8.** Plots of the mode functions $\mu_k(y)$, $\mu_k^N(y)$, and $\mu_k^E(y)$ and their relative differences $\delta^N(y)$, $\delta^E(y)$, and $\epsilon(y)$ for $k = 5$, $\beta = 5.1$, and $q_0 = 1/2$.

### 3.3. $\beta^2 \gg k^2$

In this case, two real turning points appear, given, respectively, by

$$y_1 = -y_2 = -\cosh^{-1}\left(\frac{\beta^2}{k^2}\right). \tag{64}$$

Then, we find that Equations (59) and (60) still hold in the current case, while Equation (61) is replaced by

$$\mathscr{T}(y) \to \frac{q_0^2 - 1/4}{2\beta} \ln\left(\frac{1+\epsilon}{1-\epsilon}\right) + \frac{4 + \epsilon - 5\epsilon^2}{24k(1-\epsilon^2)}, \tag{65}$$

as $y \to \infty$, but now we have $\epsilon \equiv k/\beta$. Combining Equations (59), (60) and (65), we find that currently the proper choice of $q_0$ is still that given by $q_0 = 1/2$, as in the last two subcases.

In Figure 9, we plot the quantities $|q/g|$, $|q(y-y_1)/g|$, $|q(y-y_1)(y-y_2)/g|$, and the error control function $\mathcal{T}$ for $k = 0.6$, $\beta = 4.0$, and $q_0 = 1/2$, for which we have $y_1 = -y_2 \simeq -2.58459$. From this figure, we can see that the preconditions (20)–(22) are well-satisfied. Then, for the first-order approximation of the UAA method, the solution can be approximated by Equation (34), where $\zeta_0^2$ is given by Equation (59), $\alpha_k$ and $\beta_k$ are two integration constants, and $\epsilon_1$ and $\epsilon_2$ are the errors of the corresponding approximate solutions, whose upper bounds are given by Equations (A1) and (A2) in Appendix A.

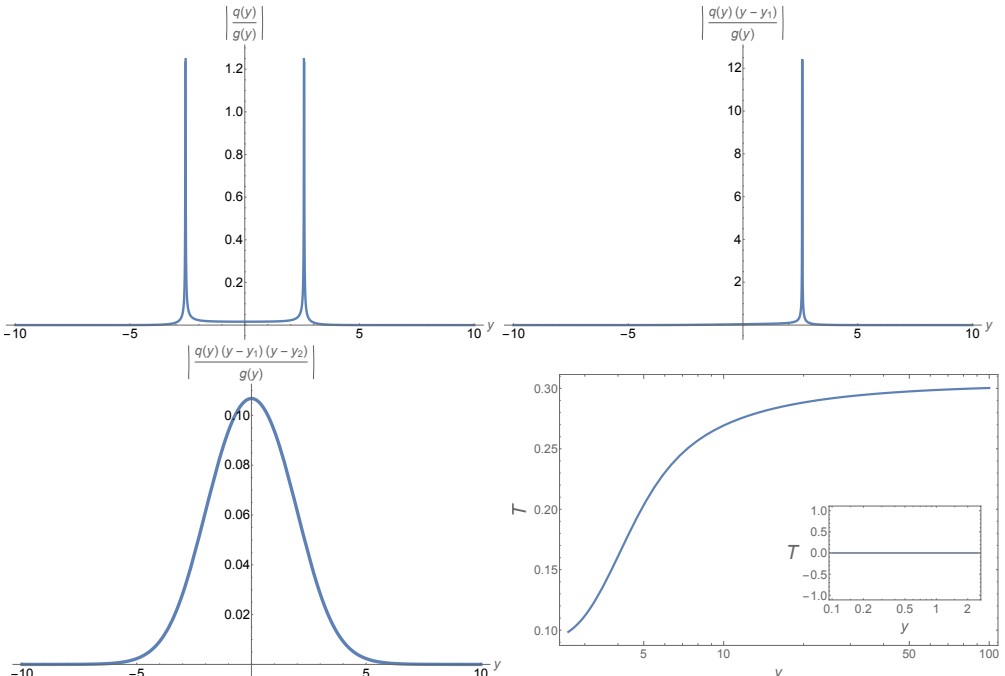

**Figure 9.** Plots of the quantities $|q/g|$, $|q(y-y_1)/g|$, $|q(y-y_1)(y-y_2)/g|$, and the error control function $\mathcal{T}$ for $k = 0.6$, $\beta = 4.0$, and $q_0 = 1/2$, for which we have $y_1 = -y_2 \simeq -2.58459$.

In Figure 10a, we plot out our first-order approximate solution, while Figure 10b is used to compare the approximate solution with the exact one, so e plot both of them. In particular, the solid line represents the exact solution, while the red dotted line is the approximate solution. From this figure, it can be seen that except for the minimal points, the two solutions match well. However, at these extreme minimal points, they deviate significantly from each. The causes of such errors are not clear, and we hope to come back to this issue on another occasion.

Finally, similar to all other cases, our numerical solution still matches well with the exact one, as shown by the curve of $\epsilon(y)$, which is no larger than $8.0 \times 10^{-6}$.

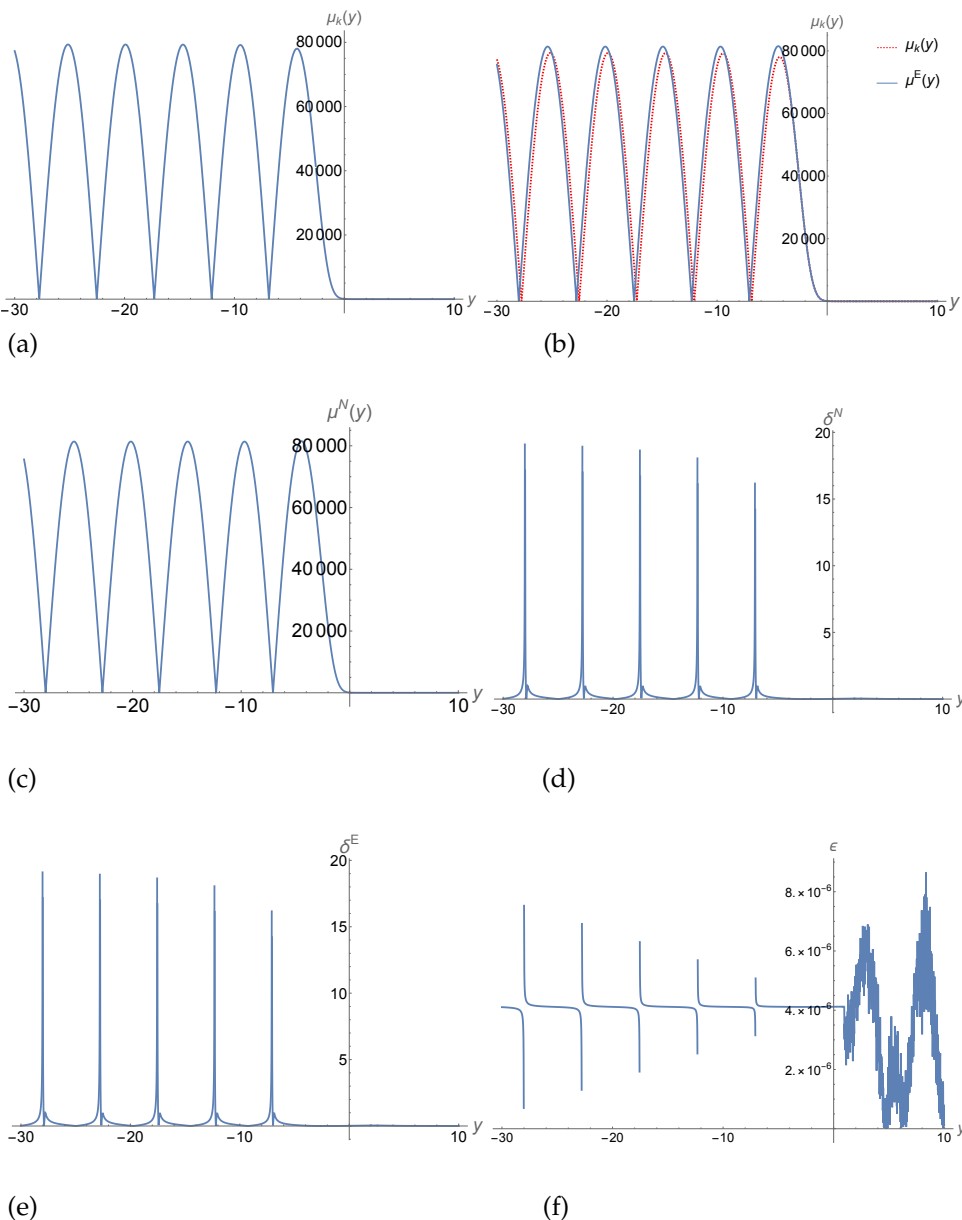

**Figure 10.** Plots of the mode functions (**a**) $\mu_k(y)$, (**c**) $\mu_k^N(y)$, and (**b**) $\mu_k^E(y)$ and their relative differences (**d**) $\delta^N(y)$, (**e**) $\delta^E(y)$, and (**f**) $\epsilon(y)$ for $k = 0.6$, $\beta = 4.0$, and $q_0 = 1/2$, for which we have $y_1 = -y_2 \simeq -2.58459$.

## 4. Conclusions

In this paper, we have applied the UAA method to the mode function $\mu_k$ with a PT potential, for which it satisfies the second-order differential equation

$$\frac{d^2\mu_k(y)}{dy^2} + \left( k^2 - \frac{\beta_0^2}{\cosh y} \right) \mu_k(y) = 0, \tag{66}$$

where $k$ and $\beta_0$ are real constants. In this case, the exact solution is known and is given by Equation (A7). The implementation of the UAA method includes the introduction of an auxiliary function $q(y)$, which is taken as

$$q(y) = \frac{q_0^2}{\cosh y}, \tag{67}$$

where $q_0$ is a free parameter. Then, we carry out the integration of the error control function, defined by

$$\mathscr{T}(\zeta) \quad \equiv \quad -\int \frac{\psi(\zeta)}{|\dot{y}^2 g|^{1/2}} d\zeta, \tag{68}$$

where

$$
\begin{aligned}
\psi(\zeta) &\equiv \dot{y}^2 q + \dot{y}^{1/2} \frac{d^2}{d\zeta^2}\left(\dot{y}^{-1/2}\right), \\
\dot{y}^2 g &= \zeta_0^2 - \zeta^2.
\end{aligned}
\tag{69}
$$

Clearly, the error control function $\mathscr{T}(\zeta)$ will depend on $q_0$. After working out the details, we find that it is convenient to distinguish between three cases: A) $k^2 \gg \beta^2$, B) $k^2 \simeq \beta^2$, and C) $k^2 \ll \beta^2$, where $\beta^2 \equiv \beta_0^2 - q_0^2 > 0$. In particular, in Case A), a proper choice of $q_0$ is $q_0 = 1/\sqrt{24}$, while in Cases B) and C), it is $q_0 = 1/2$.

Once $q_0$ is fixed, the analytical approximate solutions are uniquely determined by the linear combination of the two parabolic cylinder functions $W(\zeta_0^2/2, \pm\sqrt{2}\zeta)$, as shown by Equation (34). In particular, in Case A), the upper bounds of errors are $\lesssim 0.15\%$, as shown in Figure 4. In Case B), two subcases are considered: one with $k \gtrsim \beta$ and the other with $k \lesssim \beta$. In the first case, the upper bounds of errors are $\lesssim 4\%$, while in the second case, they are $\lesssim 10\%$, as shown, respectively, in Figures 6 and 8. In Case C), the approximate solutions also trace very well to the exact one, except for the minimal points, as shown in Figure 10. This might be caused by the fact that at these points, the mode function $\mu_k$ is almost zero, and very small non-zero values will cause significantly deviations. We are still working on this case and hope to come back to this point on another occasion.

As mentioned in the Introduction, the potentials of the mode functions in both dressed metric and hybrid approaches can be well-modeled by PT potentials. Therefore, the current analysis of the choice of the function $q(y)$ and the minimization of the error control function shall shed great light on how to carry out similar analyses in order to obtain more accurate approximate solutions in these models. We have been working on it recently and wish to report our results soon on another occasion.

In addition, the differential equations for the quasi-normal modes of black holes usually also take the form of Equation (7) with potentials that have no singularities [3], but normally do have turning points [90,91]. For example, the effective potential for the axial perturbations of the Schwarzschild black hole is given by

$$\mathscr{V}(r) = \frac{r - 2m}{r^4}\left\{l(l+1)r - 6m\right\}, \tag{70}$$

where $\omega$ denotes the quasi-normal mode. Clearly, for $r \geq 2m$, this potential also has no poles, but in general $f(r) \equiv \mathscr{V}(r) - \omega^2$ has two turning points. From [90,91], it can be seen that the properties of this potential are shared by many other cases, including those from modified theories of gravity. Thus, one can equally apply the analysis presented here to the studies of quasi-normal modes of black holes.

**Author Contributions:** All the authors have the equal contributions. All authors have read and agreed to the published version of the manuscript.

**Funding:** RP and AW were partially supported by the US Natural Science Foundation (NSF) under grant No. PHY2308845, and JJM and JS were supported through the Baylor Physics graduate program. BFL and TZ were supported in part by the National Key Research and Development Program of China under grant No. 2020YFC2201503; the National Natural Science Foundation of China under grant Nos. 11975203, 11675143, 12205254, 12275238, and 12005186; the Zhejiang Provincial Natural Science Foundation of China under grant Nos. LR21A050001 and LY20A050002; and the Fundamental Research Funds for the Provincial Universities of Zhejiang in China under grant No. RF-A2019015.

**Data Availability Statement:** No additional data are needed.

**Acknowledgments:** We thank the Special Issue Editor for his kind invitation and organization.

**Conflicts of Interest:** The authors declare no conflict of interest.

## Appendix A. Upper Bounds of Errors

The upper bounds of the errors $\epsilon_1$ and $\epsilon_2$ appearing in Equation (33) are given by

$$\frac{|\epsilon_1|}{M\left(\frac{1}{2}\lambda\zeta_0^2, \sqrt{2\lambda}\zeta\right)}, \quad \frac{|\partial\epsilon_1/\partial\zeta|}{\sqrt{2}N\left(\frac{1}{2}\lambda\zeta_0^2, \sqrt{2\lambda}\zeta\right)} \leq \frac{\kappa}{\lambda_0 E\left(\frac{1}{2}\lambda\zeta_0^2, \sqrt{2\lambda}\zeta\right)}$$
$$\times \left\{ \exp\left(\lambda\mathscr{V}_{\zeta,\zeta_2}(\mathscr{T})\right) - 1 \right\}.$$

$$\frac{|\epsilon_2|}{M\left(\frac{1}{2}\lambda\zeta_0^2, \sqrt{2\lambda}\zeta\right)}, \quad \frac{|\partial\epsilon_2/\partial\zeta|}{\sqrt{2}N\left(\frac{1}{2}\lambda\zeta_0^2, \sqrt{2\lambda}\zeta\right)} \leq \frac{\kappa E\left(\frac{1}{2}\lambda\zeta_0^2, \sqrt{2\lambda}\zeta\right)}{\lambda}$$
$$\times \left\{ \exp\left(\lambda_0 \mathscr{V}_{0,\zeta}(\mathscr{T})\right) - 1 \right\}, \tag{A1}$$

where $M\left(\frac{1}{2}\lambda\zeta_0^2, \sqrt{2\lambda}\zeta\right)$, $N\left(\frac{1}{2}\lambda\zeta_0^2, \sqrt{2\lambda}\zeta\right)$, and $E\left(\frac{1}{2}\lambda\zeta_0^2, \sqrt{2\lambda}\zeta\right)$ are auxiliary functions of the parabolic cylinder functions defined explicitly in [64], and [4]

$$\mathscr{V}_{\zeta_1,\zeta_2} \equiv \int_{\zeta_1}^{\zeta_2} \frac{|\psi(\zeta)|}{\sqrt{|\zeta^2 - \zeta_0^2|}} d\zeta, \tag{A2}$$

is *the associated error control function.*

## Appendix B. Exact Solutions with the Pöschl–Teller Potential

Let us consider the case with the Pöschl–Teller Potential given by

$$\left(\lambda^2 g + q\right) = -\left(k^2 - \frac{\beta_0^2}{\cosh^2 y}\right). \tag{A3}$$

Then, introducing the two new variables $x$ and $\mathcal{Y}$ via the relations

$$x = \frac{1}{1 + e^{-2y}}, \quad \mathcal{Y}(x) = [x(1-x)]^{ik/2}\mu_k, \tag{A4}$$

we find that Equation (7) with the above PT potential reads

$$x(1-x)\frac{d^2\mathcal{Y}}{dx^2} + [a_3 - (a_1 + a_2 + 1)x]\frac{d\mathcal{Y}}{dx} - a_1 a_2 \mathcal{Y} = 0, \tag{A5}$$

where

$$\begin{aligned} a_1 &= \frac{1}{2}(1 + \sqrt{1 - 4\beta_0^2}) - ik, \\ a_2 &= \frac{1}{2}(1 - \sqrt{1 - 4\beta_0^2}) - ik, \\ a_3 &= 1 - ik. \end{aligned} \tag{A6}$$

Equation (A5) is the standard hypergeometric equation and has the general solution [87]

$$
\begin{aligned}
\mu_k^{\mathrm{E}}(\eta) \;=\; & a_k \left( \frac{x}{1-x} \right)^{ik/2} \\
& \times \; {}_2F_1(a_1 - a_3 + 1, a_2 - a_3 + 1, 2 - a_3, x) \\
& + \frac{b_k}{[x(1-x)]^{ik/2}} \; {}_2F_1(a_1, a_2, a_3, x).
\end{aligned}
\tag{A7}
$$

Here ${}_2F_1(a_1, a_2, a_3, x)$ denotes the hypergeometric function, and $a_k$ and $b_k$ are two independent integration constants that are uniquely determined by the initial conditions.

**Appendix C. Computing the Error Control Function**

In this appendix, we collect some useful formulae for working out the error control function explicitly. Considering the particular form of the PT potential, it is easier to compute the error control function by using the new variable $x = \mathrm{sech}(y)$; thus,

$$
dy = -\frac{\epsilon_y \, dx}{x\sqrt{1-x^2}},
\tag{A8}
$$

where $\epsilon_y$ denotes the sign of $y$. In terms of the new variable,

$$
q = q_0^2 x^2, \qquad g = \beta^2 x^2 - k^2.
\tag{A9}
$$

To calculate the error control function explicitly, let us consider the cases $g < 0$ and $g > 0$ separately.

*Appendix C.1. $g < 0$*

In this case, the error control function is defined by Equation (35), which can be written as

$$
\mathscr{T}(\zeta) = \mathscr{T}_1(\zeta) + \mathscr{T}_2(\zeta) + \mathscr{T}_3(\zeta),
\tag{A10}
$$

where

$$
\begin{aligned}
\mathscr{T}_1 &\equiv \int \frac{q}{\sqrt{-g}} dy = -q_0^2 \epsilon_y \int \frac{x\,dx}{\sqrt{1-x^2}\sqrt{k^2 - \beta^2 x^2}} \\
&= \frac{q_0^2 \epsilon_y}{\beta} \ln\left( \frac{\sqrt{1-x^2}\beta + \sqrt{k^2 - \beta^2 x^2}}{\sqrt{|k^2 - \beta^2|}} \right), \\
\mathscr{T}_2 &\equiv \int \left( \frac{5 g'^2}{16 g^3} - \frac{g''}{4 g^2} \right) \sqrt{-g}\, dy \\
&= \epsilon_y \int dx \left( \frac{5\beta^4 (x^3 - x^5)}{4\sqrt{1-x^2}(k^2 - \beta^2 x^2)^{5/2}} + \frac{\beta^2 (2x - 3x^3)}{2\sqrt{1-x^2}(k^2 - \beta^2 x^2)^{3/2}} \right) \\
&= \epsilon_y \left\{ -\frac{1}{4\beta} \ln\left( \frac{\sqrt{1-x^2}\beta + \sqrt{k^2 - x^2\beta^2}}{\sqrt{|k^2 - \beta^2|}} \right) + \frac{\sqrt{1-x^2}\,A}{12(k^2 - \beta^2)(k^2 - \beta^2 x^2)^{3/2}} \right\}, \\
\mathscr{T}_3 &\equiv \int^{\zeta} \left\{ \frac{5\zeta_0^2}{4(v^2 - \zeta_0^2)^{5/2}} + \frac{3}{4(v^2 - \zeta_0^2)^{3/2}} \right\} dv = \frac{\zeta^3 - 6\zeta\zeta_0^2}{12\zeta_0^2 (\zeta^2 - \zeta_0^2)^{3/2}},
\end{aligned}
\tag{A11}
$$

where

$$
A(x) \equiv 3k^4 + 2k^2\beta^2 \left( x^2 - 1 \right) - 3x^2\beta^4.
\tag{A12}
$$

*Appendix C.2. $g > 0$*

In this case, the error control function is defined by Equation (36), which can be also written as Equation (A10) but now with

$$
\begin{aligned}
\mathscr{T}_1 &\equiv \int \frac{q}{\sqrt{g}} dy = \epsilon_y \frac{q_0^2}{\beta} \arcsin\left(\frac{\beta\sqrt{1-x^2}}{\sqrt{\beta^2-k^2}}\right), \\
\mathscr{T}_2 &\equiv \int \left(-\frac{5g'^2}{16g^3} + \frac{g''}{4g^2}\right)\sqrt{g}\,dy, \\
&= \epsilon_y \int dx \left(\frac{5\beta^4(x^3-x^5)}{4\sqrt{1-x^2}(\beta^2x^2-k^2)^{5/2}} - \frac{\beta^2(2x-3x^3)}{2\sqrt{1-x^2}(\beta^2x^2-k^2)^{3/2}}\right) \\
&= \epsilon_y \left\{-\frac{1}{4\beta}\arcsin\left(\frac{\sqrt{1-x^2}\beta}{\sqrt{\beta^2-k^2}}\right) + \frac{\sqrt{1-x^2}A}{12(\beta^2-k^2)(\beta^2x^2-k^2)^{3/2}}\right\}, \\
\mathscr{T}_3 &\equiv \int^\zeta \left\{\frac{5\zeta_0^2}{4(\zeta_0^2-v^2)^{5/2}} - \frac{3}{4(\zeta_0^2-v^2)^{3/2}}\right\}dv = \frac{6\zeta\zeta_0^2-\zeta^3}{12\zeta_0^2(\zeta_0^2-\zeta^2)^{3/2}}. \quad\text{(A13)}
\end{aligned}
$$

## Notes

[1] It should be noted that the first application of the UAA method to cosmological perturbations in GR was carried out by Habib et al.

[2] In this case, the associated error control function is $\mathscr{V}_{\zeta_1,\zeta}(\mathscr{T})$ for any given $\zeta_1$, where $\zeta_1 \in (-\infty, \infty)$ [64]. In this paper, we choose $\zeta_1 = 0$, so the integrations will be carried out over the interval $\zeta \in [0, \infty)$, corresponding to $y \in [0, \infty)$. Due to the symmetry of the equation, one can easily obtain the solutions for the region $y \in (-\infty, 0]$ by simply replacing $y$ by $-y$ (or $\zeta$ by $-\zeta$)

[3] recall the inner boundaries of black hole perturbations are the horizons, at which the potentials are usually finite and non-singular

[4] This corresponds to choosing the function $\Omega(x)$ introduced by Olver in [64] as $\Omega(x) = \sqrt{|x^2 - \zeta_0^2|}$, which satisfies the requirement $\Omega(x) = \mathcal{O}(x)$ as $x \to \pm\infty$. For more details, see [64].

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
