# Peer review of "Uniform Asymptotic Approximation Method with Pöschl–Teller Potential"

_universe, doi:10.3390/universe9110471_

Round 1

Reviewer 1 Report

Comments and Suggestions for Authors

In this paper, the authors study analytical approximate solutions of the second-order homogeneous 1 differential equations with the existence of only two turning points (but without poles), by using 2 the uniform asymptotic approximation (UAA) method.  The author used the PT potential as an example, showing that this method is good for solving the Schrodinger-like equation. This work also indicates a great potential for obtaining QNMs in black hole physics, and further Gravitational wave investigations. Based on this, I recommend this manuscript be published. 

Author Response

Thanks a lot for the positive report, and i really appreciate the support. 

Reviewer 2 Report

Comments and Suggestions for Authors

In this paper the authors describe approximate solutions of the second order homogeneous differential equations with two turning points, in particular they use the Poschl-Terrel potential for which analytical solutions are known. The authors perform a thorough study on the properties of the equation and their solutions. 

The mathematical approach is quite elaborated but clear to follow, however I would recommend the authors provide a physical system in which the results can be applied due the nature of the Journal. For instance, they say by the end of the manuscript the analysis they developed can be used to calculate qnm of black holes. This will also help to get a better connection between the physical motivation given in the Introduction with the rest of the manuscript.

Author Response

Following the referee's comments, at the end of Conclusion, we have added some sentences to show that QNMs of black holes indeed have the properties of the potential considered in the current paper, that is, with two turning points and no poles. So, our method can be easily applied to the studies of QNMs of black holes. We have also added the potentials studied in LQC in the dressed metric and hybrid approaches, and show how they are related to the PT potential explictly. For the sake of convince of the referee, we have marked all the changes in red letters. The rest of the paper remains the same. 

Reviewer 3 Report

Comments and Suggestions for Authors

Reviewer’s report on  Manuscript  ID: Universe-2677588-

Uniform Asymptotic Approximation Method with Pöschl-Teller Potential

by R. Pan, J.J. Marchetta, J. Saeed1, G. Cleaver, B. Li, A. Wang,_ T. Zhu

 The manuscript is devoted to finding  analytical approximate solutions of the second-order homogeneous differential equations (with Pöschl-Teller  potential) with the existence of only two turning points, by using the uniform asymptotic approximation (UAA) method.

 The authors have reached interesting results with analytical solutions of the second-order homogeneous differential equations with Pöschl-Teller  potential of the form (4.1).

At the same time, the relation to the Cosmology (main direction in Universe) seems to be rather artificial. At the same time no one other than the authors will be able to apply the developed approximation technique to the problem of Cosmology (studies of quasi-normal modes of black holes). The last problem is really interesting.

 As a whole the paper contains new result for the numerical solutions  of the second-order homogeneous differential equations with Pöschl-Teller  potential with sufficient list of plots and deserves to be published in the Universe.

I think the manuscript maybe accepted for publication in the Universe but after elaboration of more concrete example which will show the application of the method to the studies of quasi-normal modes of black holes (at least on the initial stage).

Author Response

(The authors gave the same response as above.)

Round 2

Reviewer 2 Report

Comments and Suggestions for Authors

The authors performed modifications concerning the suitability of the model to compare with some results of black hole physics and I consider the manuscript is suitable for publication.